# Endothelial Dysfunction, Erectile Deficit and Cardiovascular Disease: An Overview of the Pathogenetic Links

**DOI:** 10.3390/biomedicines10081848

**Published:** 2022-08-01

**Authors:** Federico De Leonardis, Gaia Colalillo, Enrico Finazzi Agrò, Roberto Miano, Andrea Fuschi, Anastasios D. Asimakopoulos

**Affiliations:** 1Unit of Urology, Department of Surgical Sciences, University of Rome Tor Vergata, 00133 Roma, Italy; deleonardis96@gmail.com (F.D.L.); gaia_colalillo@hotmail.it (G.C.); finazzi.agro@med.uniroma2.it (E.F.A.); mianor@virgilio.it (R.M.); 2Urology Unit, Department of Medico-Surgical Sciences and Biotechnologies, Faculty of Pharmacy and Medicine, Sapienza University of Rome, 04100 Latina, Italy; andreafuschi@gmail.com; 3Unit of Urology, Fondazione PTV Policlinico Tor Vergata, 00185 Rome, Italy

**Keywords:** cardiovascular diseases, coronary artery disease, erectile dysfunction, endothelial dysfunction

## Abstract

Erectile dysfunction (ED) is a condition with multifactorial pathogenesis, quite common among men, especially those above 60 years old. A vascular etiology is the most common cause. The interaction between chronic inflammation, androgens, and cardiovascular risk factors determines macroscopically invisible alterations such as endothelial dysfunction and subsequent atherosclerosis and flow-limiting stenosis that affects both penile and coronary arteries. Thus, ED and cardiovascular disease (CVD) should be considered two different manifestations of the same systemic disorder, with a shared aetiological factor being endothelial dysfunction. Moreover, the penile arteries have a smaller size compared with coronary arteries; thus, for the same level of arteriopathy, a more significant blood flow reduction will occur in erectile tissue compared with coronary circulation. As a result, ED often precedes CVD by 2–5 years, and its diagnosis offers a time window for cardiovascular risk mitigation. Growing evidence suggests, in fact, that patients presenting with ED should be investigated for CVD even if they have no symptoms. Early detection could facilitate prompt intervention and a reduction in long-term complications. In this review, we provide an overview of the pathogenetic mechanisms behind arteriogenic ED and CVD, focusing on the role of endothelial dysfunction as the common denominator of the two disorders. Developed algorithms that may help identify those patients complaining of ED who should undergo detailed cardiologic assessment and receive intensive treatment for risk factors are also analyzed.

## 1. Introduction

Erectile dysfunction (ED) in men is defined as the inability to obtain or sustain an erection satisfactory for sexual intercourse. ED prevalence is directly correlated to patients’ age. Moreover, its incidence increases if specific subgroups of the population are taken into account (smokers, diabetics, obese, and people affected by cardiovascular diseases) [1].

There are multiple mechanisms that can lead to ED and include (i) psychological distress, (ii) neurological processes impairing the correct transmission of signals and stimuli that trigger the erection (i.e., multiple sclerosis, spinal cord injuries, neuropathies, and pelvic surgeries), (iii) hormonal imbalances, and (iv) all the vascular conditions that may alter normal blood flow and oxygen supply to tissues, such as smoking, diabetes, atherosclerosis, and more generally any condition that may disrupt endothelial homeostasis.

Endothelial dysfunction, in particular, seems to share pathological mechanisms with and be the starting point of many of the aforementioned conditions, implying an interconnection between all these affections. This raises the question of whether there may be merit in ED early identification for the screening and prevention of cardiovascular disease (CVD) [2].

ED is still a topic that men feel shame talking about and seeking medical attention for, making such an important issue for men’s wellbeing probably underdiagnosed and underestimated.

There is no concrete data about ED prevalence in the global population, as different rates are encountered among various ethnicities and geographical areas, apparently being more common in North America and Southeast Asia [3,4]. Two studies analyzed ED incidence in the general male population in-depth, respectively, in Europe and the United States [5,6]. In the first study, the European Male Aging Study (EMAS), ED had a prevalence of 30% in the entire screened population, and it was directly correlated to age, surging at 64% incidence in men older than 70 yrs. As for the American study, the Massachusetts Male Aging Study (MMAS), ED of any degree was found to occur in 52% of the male population ranging from 40 to 70 years old, and once again with severity and occurrence directly related to the patients’ age. In a more recent American study [1], in addition to the well-known correlation between ED and age, the incidence of ED was shown to be even higher when specific subgroups of the population were taken into account, such as in cases of concomitant smoking (13%), diabetes (51.3%), obesity (21.8%), and cardiovascular disease (50%).

Less data are available on how ED impacts the younger population. An increase in ED incidence was demonstrated in men younger than 40 years that was again potentially underestimated due to the scarce self-reporting of younger patients to clinicians. The rates of diagnosis of organic causes for ED in younger patients is increasing, bringing down the misconception that ED in such a population would be most uniquely due to psychological causes. This emphasizes the complexity and the interconnecting links of all the possible causes of ED, showing that some of them may remain unnoticed or be difficult to identify [7,8].

## 2. Physiology of Erection and Physiopathology of Vasculogenic ED

The necessary event for the male sexual act to start is the acquisition and maintenance of penile erection, which involves an interplay between hormonal, psychological, neurological, and vascular pathways.

Erections are primarily under the control of the parasympathetic nervous system, which, via the pelvic plexus and cavernous nerve, triggers the vasodilation of the penile blood vessels, thus increasing the amount of blood in the sinusoidal spaces of corpora cavernosa and corpus spongiosum and engorging them with blood. Once the blood fills these cavities, erection is sustained by the reduced venous outflow (via the compression of subtunical venules exerted by the cavernous tunica albuginea), as well as by the relaxation of erectile tissue smooth muscle fibers [9]. While the former mechanism is a mechanical event, the latter is once again regulated by the parasympathetic nervous system.

In fact, penile smooth muscle fibers stay tonically contracted when in a flaccid state; they are activated during erection following triggers arriving from the parasympathetic nervous system and the vascular endothelium of the penile vessels. Once the parasympathetic axons are activated, the norepinephrine levels, which directly limit the release of nitric oxide (NO), drastically decrease, and two fundamental molecular mediators are discharged: acetylcholine (ACh) and NO. When released, ACh binds to M3 muscarinic receptors on endothelial cells. By doing so, these receptors will, in turn, directly either increase preformed NO release or induce a specific endothelial isoform of NO synthase (eNOS) activity, thus increasing NO production. Once released, NO activates an enzyme on the smooth muscle membrane that converts GTP to cGMP, with the latter being accountable for the phosphorylation of calcium ion channels and sarcoplasmic reticulum membrane proteins, thus causing cellular hyperpolarization and ultimately smooth muscle relaxation [10,11,12].

Given the complexity of the interplay between various pathways that cause an erection, it should be erroneous to think of ED pathophysiology in a dichotomous way (organic vs. psychological pathways). ED is a multifactorial disorder deriving from the synergy of social context, psychological wellbeing, and organic processes. Consequently, there are plenty of ways in which these linked processes can become disrupted, ultimately causing ED. 

## 3. The Endothelium: Summary of Its Functions

The endothelium is the innermost structure present inside the blood vessels, the functions of which go beyond that of having a simply structural role. In fact, endothelial cells are multifunctional cells owning several metabolic and synthetic properties. In a normal state, they play a pivotal role in maintaining circulatory and blood homeostasis by exerting a direct inhibitory effect on platelet activation [13] via the secretion of prostaglandin (PGI2), nitric oxide (NO), and adenosine diphosphatase; by synthetizing a fibrinolytic enzyme such as tissue plasminogen activator (t-PA) and by secreting anticoagulant proteins (thrombomodulin, heparin-like molecules, and tissue factor pathway inhibitors) [14].

The endothelium also regulates blood flow and vascular resistances by secreting vasoconstricting (endothelins, thromboxane A2, and angiotensin-converting enzyme (ACE)) or vasodilating (NO and prostacyclins) agents, thus regulating oxygen and metabolites supplying to different organs and tissues [15]. Furthermore, endothelium contributes to hormonal regulation and other metabolic activities by synthesizing and secreting several growth factors (mainly hemopoietic colony-stimulating factors (CSFs)) and by limiting the oxidation of lipoproteins involved in atheromatous plaque formation [16]. If the endothelial cells display all of these different functions while in their regular state, this equilibrium is totally subverted when the endothelium is damaged, inducing endothelial cell activation and expression of newly synthetized cytokines and adhesion molecules [17]. Stimuli that can induce this shift towards a pro-coagulant and pro-inflammatory status include a series of intrinsic non-modifiable risk factors, such as an abnormal blood flow (either excessively static or too turbulent), a proper thrombophilic predisposition of the patient (both primary or acquired), or modifiable risk factors, which include infections, endothelial disruption, metabolic abnormalities (homocystinemia or hypercholesterolemia) and presence of toxins (i.e., smoking) [18].

While in this activated state, the endothelium starts expressing pro-coagulant surface molecules, attracting pro-inflammatory cells and expressing factors limiting NO availability (thus causing smooth muscle and consequently vessel contraction) [19]. This cascade of events ultimately leads to the activation and proliferation of smooth muscle cells and excessive deposition of collagen and matrix, thus causing intimal thickening that may even hinder vascular flow.

## 4. Endothelial Dysfunction and ED

Endothelial dysfunction is an early-stage vascular damage, representing the concept that any kind of stress or insult acting on the vascular endothelium may hinder its physiological NO production and release, together with impairing relaxation and distension of the involved vessels (Figure 1). Many studies investigated the association between endothelial dysfunction and ED. What was actually highlighted was how this mechanism represents the starting point not only for ED pathogenesis but also for many other pathological processes, paving the way to understand how ED and CVD could be linked and influence each other.

Endothelial dysfunction implies an altered responsiveness to vasoactive stimuli (i.e., lack of eNOs availability, reduced responsiveness to NO vasodilatory action, and/or an increased response to vasoconstricting agents) [20] together with actual structural changes. The latter ones are represented by tunica intima thickening, increased endothelial permeability, or macrophage deposition in vessel walls [21].

Specifically regarding ED, within a dysfunctional endothelium, there seems to be an altered response to ACh release since it causes an altered NO production or even paradoxical vasoconstriction, as it was also observed in the coronaries of patients affected by coronary artery diseases [22]. Another mechanism contributing to an altered NO response can be attributed to free radical oxygen species that may react with NO, directly decreasing its bioavailability and originating active bioproducts that may exert oxidative damages [23]. This same mechanism also occurs in atheromatous plaque formation, which represents a fundamental risk factor for CVDs and may alter vascular smooth muscle ability to relax and distend: an essential feature for penile erection [24,25].

After having ascertained the role played by an altered NO release, it must be specified that this is just one of the many mechanisms involved in endothelial dysfunction. Other pathways worthy of mention are the ones played by the Tumor Necrosis Factor alpha (TNF-a). It is produced by endothelial and smooth muscle cells in response to increased oxidative stress or inflammatory conditions [26,27]. TNF-a induces ROS generation, which may directly lower NO levels, apart from suppressing eNOS expression [28]. Moreover, TNF-a in endothelial cells causes atherosclerosis [29] and increases junctional cell permeability and, ultimately, cell apoptosis [30]. Similar effects of altered cell permeability and increased cellular apoptosis were also described in diabetic patients when studying carbonic anhydrase (CA) activity. CAs, a family of metalloenzymes also involved in pH regulation and ion transport, were shown to play a major role in diabetic microangiopathy. In patients with high plasma glucose levels, CAs type I overexpression was responsible for altered endothelial cell permeability and consequent cellular death in vitro, which results in a decreased myocardial capillary density on histopathological specimens [31]. Simultaneously CA type II was shown to be responsible for a particular ion exchanger (NHE-1) hyperactivity, which ultimately leads to ROS generation, myocardial dysfunction, and cellular apoptosis.

## 5. ED and CVD

ED can be, at least partially, considered a vascular disease, as many of its risk factors are the same ones as for cardiovascular diseases (in particular, the presence of diabetes, hypertension, smoking, or aging). Thus, a possible connection between the two clinical entities (i.e., ED and CVD) has been suggested [32,33].

It has already been described in literature how ED and CVD are supposedly crosslinked, even if scarce data were available to define certain results [34,35]. Later on, while studying patients with overt and clinically significant cardiac diseases, ED was found to affect at least 50% of patients involved in those research, with the study by Solomon et al. uncovering the fact that most of these patients noticed ED symptoms prior to any major cardiac issue hinting to a possible sentinel role of ED in preventing CVD [36,37].

Another fundamental study that laid the groundwork for the potential utilization of ED symptoms in CAD early diagnosis is represented by the one of Montorsi et al., in which a group of 300 men affected by clinically significant coronary artery disease was examined to assess ED prevalence [38]. The results showed that almost half of patients affected by CAD experienced ED, but most importantly, it described how 65% of patients had ED symptoms before any cardiovascular event, estimating a 3-year interval between the onset of the two processes. The authors interpreted these data as representative of the “artery-size hypothesis” theory, implying that since penile vessels are smaller than coronary arteries, any pathological endothelial process causing an alteration or significant vascular flow stenosis would impact the male reproductive and vascular bed first, rather than immediately causing overt cardiac symptoms [39].

In addition to the aforementioned studies questioning the ED prevalence in CAD patients, the relation between silent CVD in ED patients has also been investigated. Pritzker et al. were the first to observe men who had ED and a silent medical history for CVD, demonstrating via medical questionnaires that 80% of these patients shared multiple cardiac risk factors. [40]. Furthermore, almost 60% of the enrolled patients failed a treadmill stress test, and among this subgroup, one to three vessel diseases were discovered in 70% of patients, suggesting the presence of silent ischemia in apparently healthy ED patients. Some of the arguably most important insights were obtained after the study conducted by Vlachopoulos et al. that prospectively investigated by angiography the incidence of CAD in 50 patients with ED and no prior relevant cardiological medical history [41]. The authors noticed that silent cardiological alterations were present in 20% of the patients. More specifically, an increased risk of roughly 45% for CV events and 25% for overall mortality in men with ED compared with those without, was shown.

Similar studies have been conducted, trying to correlate ED severity to CAD entity. A large prospective Australian study showed how ED prevalence directly increased with age and was higher among smokers and, most of all, that ED severity directly correlates with the risk of developing future cardiovascular events, despite the presence or not of any already diagnosed CVD [42]. Moreover, Montorsi et al. described in the COBRA Trial how ED severity correlates directly with CAD extension, stressing the presence of an extended multivessel disease in patients with apparently stable CAD but severe ED [43]. Similar results were obtained even when investigating coronary artery calcifications [44].

Kaiser et al. [45] analyzed instead whether patients with ED and no clinically apparent CVD had any pathological vascular abnormality other than in the penile vascular bed. The authors established via some vascular wellbeing parameters that patients who received a Doppler diagnosis of ED but had no other CVD scored significantly lower than the control group. They concluded that systemic vascular defects in endothelium vasodilatation actually take place before the development of any other overt functional or structural systemic vascular disease.

## 6. ED Therapeutic Approach

Another correlation between ED and CVDs was also noted regarding the effects of a healthy lifestyle or appropriate medical therapy, which may impact the general prognosis of people affected by ED. The rationale for this is that phosphodiesterase 5 inhibitors (PDE5-Is), the first class of drugs used to treat ED, are able to increase NO vascular production by inhibiting the enzyme supposed to degrade cGMP, thus eliciting an erection by the aforementioned mechanisms. What is needed for these drugs to work, apart from an adequate sexual stimulus, is a correctly functioning system of NO release from the nervous fibers and corpora cavernosa endothelium. Among the various types of PDE5-Is (mainly Sildenafil, Tadalafil, Avanafil, and Vardenafil), no significant differences in efficacy were noted, making the selectivity and tolerability profiles of each drug the main discriminants when starting a therapy. These drugs exert a systemic effect, potentially bringing benefits to the stenotic coronary arteries. It was described how PDE5-Is administration was able to reduce the risk of cardiac events in diabetic patients with not yet clinically significant CAD and decreased levels of systemic proinflammatory markers, where chronic inflammation is supposed to play an important role in both ED and CVD pathogenesis [46]. Moreover, although large and numerous clinical trials in humans are missing, the implementation of nutraceuticals before starting any type of treatment, or eventually in addition to it, may represent an important resource for ED prevention and the maintenance of vascular health since they may promote endothelial vasorelaxation, exert antioxidant properties, and modulate the lipidic profile. Furthermore, the patient’s potential predisposition to start natural-substances-based therapy could represent the best way to increase a patient’s compliance to a preventive strategy and, thus, potentially reduce the incidence of CVDs. In the end, it is reasonable to consider the use of nutraceuticals as complementary to the “traditional” lifestyle changes and “common” pharmacological therapies in order to improve the patient’s outcome [47,48,49].

Although some results could seem promising, more data about PDE5-Is effects in patients with a cardiovascular impairment need to be obtained, as NO deficit is just one of the many actual mechanisms behind ED and CVDs. In fact, altered NO production due to endothelial dysfunction just one of the potential causes of ED; it should be reasonably feasible to treat ED indirectly by treating its risk factors or predisposing conditions. New therapeutic approaches are being investigated; in particular, oral hypoglycemic agents are among the ones showing the most promising results. It was already shown how, in patients affected by diabetes, specific medical treatments also helped in a lot of conditions that could be brought back to the primary disease. Attention has been focused on drugs that could also provide benefits to non-diabetic patients regardless of their glucose levels, hinting at the presence of multiple mechanisms leading to ED. Metformin, for example, was demonstrated to be able to reduce vascular inflammation, atherosclerosis, and oxidative damage, regardless of changes in body weight or glycated hemoglobin [50]. The employment of Gliflozins (another class of hypoglycemic agents) instead was correlated to a general improvement in endothelial function, probably due to reduced glucotoxicity, change in lipidic profile, anti-inflammatory action, and a possibly enhanced induction of vasodilation [51,52].

## 7. Cardiologic Assessment Recommendations for Patients Affected by ED

It is still unclear whether all men affected by ED should be screened for any cardiovascular damage risk or even actual disease, but more evidence is always being published about this topic. As already stated, ED is supposed to precede CVDs onset by roughly 3 years, giving us the opportunity to properly screen or treat our patients before any major cardiac event takes place. In fact, it was demonstrated by Vastergaard et al. that men with no overt prior CVD and that received treatment for ED showed a decreased risk for acute myocardial infarction compared to the general male population, especially in the first year after the start of treatment for ED [53]. The risk of heart failure slightly decreased in the 3 years after initiation of ED medication as well; however, no significant difference was evidenced regarding stroke incidence.

In the various attempts to stratify men affected by ED for the risk of developing CVDs, a valid contribution was provided by the Third Princeton Consensus conference [54]. These experts focused on re-evaluating the cardiac risk associated with sexual activity in patients with already overt CVDs, and mostly on identifying men with ED who could benefit from a cardiological expert evaluation. Even though an ED assessment has been included in the United Kingdom QRISK-3 calculator (a prediction algorithm for CVD), it is still neglected by major CVDs risk scores [55]. The usual scale used to assess cardiovascular risk is the Framingham Risk Score (FRS). The Consensus suggests that the FRS evaluation be used as a first-step tool for estimating the likelihood of subclinical cardiovascular dysfunction in all men with ED. This scoring system uses parameters considered to be the most important CVD risk factors, including age, sex, cigarette smoking, systolic blood pressure, use of blood pressure medications, total cholesterol, and high-density lipoprotein (HDL) cholesterol in an attempt to estimate an individual’s 10-year risk for cardiovascular events of any kind. However, this score presents some pitfalls, as it probably underestimates the risk in younger patients, who would instead benefit from a longer time span assessment [56]. Moreover, its validity in European and nonwhite populations has been questioned, and it was pointed out how this scoring system does not even take into account some important factors such as the patient’s family history, kidney function, or testosterone levels, i.e., all well-known risk factors for CVDs [57,58].

According to the available evidence, the addition of the ED parameters to FRS caused some patients to be re-classified in a higher risk category [59]. Furthermore, it was described how the presence of organic ED in patients <40 years strictly correlates with an increased incidence of atherosclerotic cardiovascular events [58,59]. The severity of ED (assessed via the international index of erectile function (IIEF) or Sexual Health Inventory of Men (SHIM) scores) was also indicative of the potential CVD damage [60]. In an observational study conducted by Fang, it was investigated the association between ED status and the Framingham CVDs risk group, as well as if any change in ED status would be accompanied by a Framingham risk modification [61]. What the authors described was that ED, regardless of it being transient or not, was significantly associated with an increased Framingham risk. In particular, when changes in the Framingham risk over time were considered, the results came out significant only in the younger population, while no associations were observed in older men, probably due to the higher baseline CVDs risk in this latter category.

According to the American College of Cardiology Foundation, while low-risk patients should only initially undergo noninvasive tests, intermediate-risk patients could benefit from performing more accurate (but more invasive) diagnostic techniques, such as coronary computed tomographic angiography (CCTA) and coronary artery calcium scoring (CACS). These tests, nonetheless, could immediately represent the first test of choice when studying patients with a strong family history of CVDs, with several risk factors, or with an unusual clinical presentation (i.e., young patients). For patients with organic ED and with no prior cardiological exam available, an accurate evaluation for CVDs should be conducted via noninvasive methods at first, reserving more invasive maneuvers only for those patients who are confirmed in the high-risk class.

A noninvasive cardiovascular evaluation includes stress tests for ischemia and serum biomarkers measurements. The latter consists of assessing the levels of serum molecules, which could represent a new cost-effective method to evaluate the correct risk category for each patient. The serum biomarker with the most promising results seems to be the high sensitivity C Reactive Protein (hsCRP), being probably representative of the underlying status of chronic inflammation characterizing endothelial dysfunction and thus both ED and CVDs [62]. In a study conducted by Ridker, it was described how hsCRP measurements caused a significant redistribution of patients at intermediate risk for CVDs into higher or lower risk groups and how it could also be a predictor of the potential benefit obtained with statin therapy [63]. Moreover, Ferrandis-Cortes studied how patients with hsCRP > 1 mg/L, considered to have a moderate/high cardiovascular risk, showed significantly lower testosterone levels and significantly higher diabetes-related variables (i.e., BMI or waist circumference), all established risk factors for CVDs [64].

Another serum marker that should be taken into account while treating patients with ED is testosterone. Although testosterone measurements are not always performed when studying a patient with ED, it was demonstrated in the prospective study conducted by Laughlin how increasingly lower testosterone levels were associated with a parallel increase in cardiovascular diseases and all-cause mortality [65]. Similar results were obtained by Haring and his collaborators, this time also introducing the idea that low testosterone may not be a direct risk factor but instead a risk marker for CVDs and all its predisposing conditions (such as diabetes or metabolic syndrome) [66].

In particular, testosterone should be investigated in all those patients with ED where therapy with PDE5-I failed in order to set up the best feasible therapy [67]. However, although low testosterone levels are associated with an increase in mortality due to cardiovascular events, testosterone replacement therapy did not show a substantial reduction in the incidence of CVDs in men affected by ED [68]. What was described instead was that it could bring symptomatic benefits to men with ED and chronic heart failure without nonetheless an associated increase in cardiovascular risk [69].

Even if patients first come to our attention for ED, treatment should actually be postponed until a proper cardiological evaluation has been made, as it may influence the correct therapeutic approach. Vlachopoulos et al. proposed, based on the major prognostic cardiovascular scores and the presence or not of an already overt cardiological impairment or diabetes, an algorithm that tries to define the best management for patients affected by ED according to their cardiovascular risk group [70]. More specifically, they also took into account the possibility of the occurrence of a new cardiovascular event during sexual intercourse since, in a metanalysis conducted by Dahabreh, acute cardiac events came out to be significantly associated with episodic physical and sexual activity [71]. These results were already introduced by Drory et al., who were able to demonstrate asymptomatic ST segment depression during sexual intercourse via electrocardiographic monitoring in over 30% of patients with known coronary artery disease enrolled in their study [72]. Energetic demand during sexual intercourse has been established to account for roughly 3 Metabolic Equivalent Tasks (MET), the same as walking at a speed of 3 km/h; thus not excessively energy consuming but comparable to an everyday physical effort [73]. So, according to Vlachopoulos et al. the low-risk group should include otherwise healthy patients, for example, those with controlled hypertension and who are able to achieve the goal of 5 MET during a stress test, as well as those who recovered completely from myocardial infarction [72]. These patients have no contraindication to sexual activity; therefore, starting ED therapy with PDE5-I is encouraged without any major concern. There are, however, a few things to remember when dealing with low-risk patients. Antihypertensive medications, especially beta blockers and thiazide diuretics, can contribute to sexual dysfunction [74], and since changing medications or dosages may not solve the problem, men with controlled hypertension may need therapy acting directly against ED. Moreover, those men taking nitrates for stable angina need to have their treatment scheme modified if they are willing to take a drug for ED, as nitrates are an absolute contraindication to PDE5-I use due to potential severe hypotension.

Intermediate-risk patients include diabetics, men with congestive heart failure, or who experienced myocardial infarction 2 to 8 weeks ago. These patients, even according to the Princeton panel, have an uncertain cardiac condition and should therefore be reassessed and reclassified either in the high- or low-risk class after an appropriate evaluation. For them, a stress test is usually used to properly allocate them into the correct risk class. Recently, in this subset of patients, six potential CVD markers (hsCRP, family history for CVDs, CACS, ankle-brachial index, carotid intima-media thickness, and brachial flow-mediated dilation) were compared to improve the prediction of cardiovascular events [75]. Among these biomarkers, CACS yielded better results, but without being able to identify a correct sequence of exams to perform or to prove certain diagnostic superiority of one test over another. Therefore, in intermediate-risk patients, while lifestyle modifications should begin promptly, it is suggested to postpone ED treatment until a stable cardiac condition is reached [76,77]. Lastly, in the high-risk group can be found patients with unstable or refractory angina, uncontrolled hypertension, high-risk arrhythmias, and those who suffered from myocardial infarction within 2 weeks. For these patients, the cardiac condition is so severe that sexual intercourse may represent a serious risk for an acute cardiovascular event, and they should therefore postpone any sexual activity and ED treatment after a proper and accurate cardiological evaluation.

This is just one of the potential algorithms that take ED and sexual activity into account that could be used when assessing CVD risk. A global approach aimed at reducing cardiovascular risk (either with pharmacological treatment or lifestyle changes) actually improves overall cardiovascular health, including sexual function. All of these recommendations should help the physician, no matter their specialty, to properly assess the general health status of patients and correctly treat them. Men with sexual dysfunction are frailer than initially thought, and ED is a red flag of more serious issues.

## 8. Conclusions

ED is a multifactorial disorder with a really intricated pathogenesis. Vasculogenic ED and CVD should be considered as two different representations of the same disorder since they share common risk factors and pathophysiological mechanisms, with the main being endothelial dysfunction. 

Screening and diagnosing ED is essential for the primary and secondary prevention of CVDs, as the assessment of ED represents an easy and low-cost prognostic tool if compared to various cardiovascular biomarkers or more invasive exams. For men believed to have predominantly vasculogenic erectile dysfunction, initial cardiovascular risk stratification should be performed, for example, applying the Framingham Risk Score. The management of men with erectile dysfunction who are at low risk for cardiovascular disease should focus on risk-factor control, whereas men at high risk, mostly including those with active cardiovascular symptoms, should be referred to a cardiologist before starting any kind of therapy for ED. Intermediate-risk men should undergo noninvasive evaluation for subclinical atherosclerosis and usually be re-stratified in any of the two previous groups. In conclusion, the penile vascular bed could be a sensitive indicator of systemic vascular disease, making ED the herald of major cardiovascular events. With the latter being the leading cause of morbidity and mortality in Western countries, cardiovascular risk stratification and risk-factor management should be promptly suggested in all men with vasculogenic erectile dysfunction. Further efforts to raise awareness about the potential role of ED in the early diagnosis and prevention of CVDs are required.

## Figures and Tables

**Figure 1 biomedicines-10-01848-f001:**
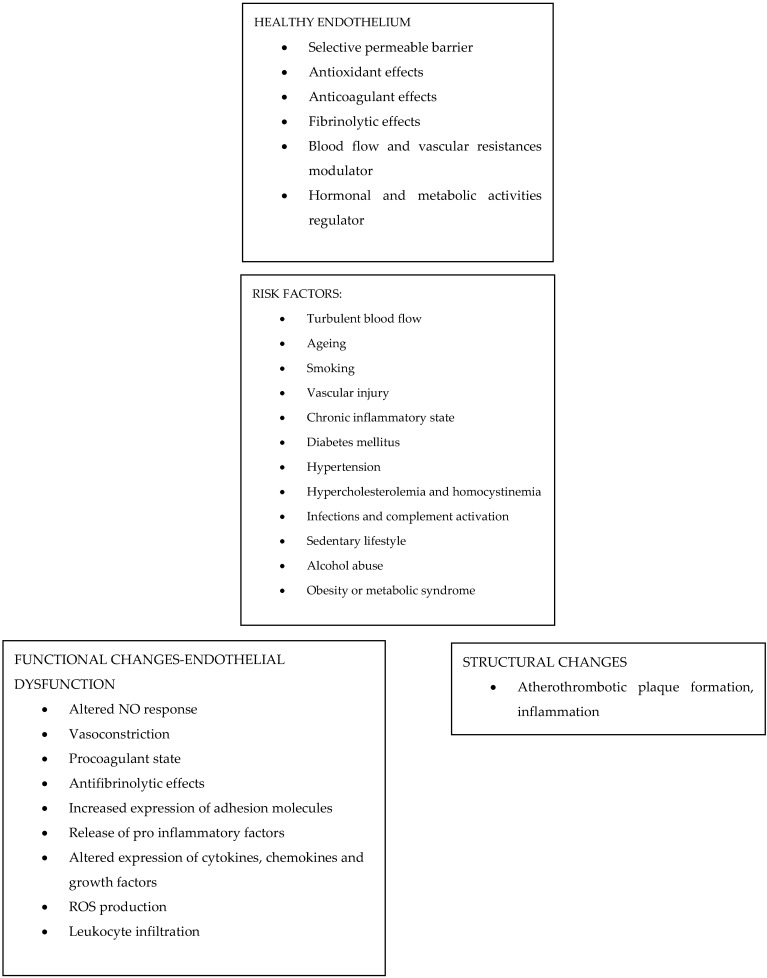
Endothelial Dysfunction and ED.

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
