# Peer review of "Endothelial Dysfunction, Erectile Deficit and Cardiovascular Disease: An Overview of the Pathogenetic Links"

_biomedicines, 2022, doi:10.3390/biomedicines10081848_

Round 1

Reviewer 1 Report

The review highlights the role of endothelial dysfunction in erectile deficit and CVD. The review touches an interesting argument on an often under diagnosed problem. The article is well written, simple to read and accessible to a broad audience. 

Before publication minor adjustments are required:

-I would include a section for disease treatment, highlighting the use of nutraceuticals active on oxidative stress. (Carrizzo, Albino et al. “A Novel Promising Frontier for Human Health: The Beneficial Effects of Nutraceuticals in Cardiovascular Diseases.” International journal of molecular sciences vol. 21,22 8706. 18 Nov. 2020, doi:10.3390/ijms21228706 ; Carrizzo, Albino et al. “New Nutraceutical Combination Reduces Blood Pressure and Improves Exercise Capacity in Hypertensive Patients Via a Nitric Oxide-Dependent Mechanism.” Journal of the American Heart Association vol. 9,5 (2020): e014923. doi:10.1161/JAHA.119.014923 ; Di Pietro, Paola et al. “A Novel Combination of High-Load Omega-3 Lysine Complex (AvailOm®) and Anthocyanins Exerts Beneficial Cardiovascular Effects.” Antioxidants (Basel, Switzerland) vol. 11,5 896. 30 Apr. 2022, doi:10.3390/antiox11050896 )

- The conclusion would benefit from added considerations. What should future studies search for in ED from Endothelial dysfunction and what do the authors think should the clinical recommendations suggest in patients with ED at risk for CVD. 

Author Response

The review highlights the role of endothelial dysfunction in erectile deficit and CVD. The review touches an interesting argument on an often under diagnosed problem. The article is well written, simple to read and accessible to a broad audience. 

Before publication minor adjustments are required:

-I would include a section for disease treatment, highlighting the use of nutraceuticals active on oxidative stress. (Carrizzo, Albino et al. “A Novel Promising Frontier for Human Health: The Beneficial Effects of Nutraceuticals in Cardiovascular Diseases.” International journal of molecular sciences vol. 21,22 8706. 18 Nov. 2020, doi:10.3390/ijms21228706 ; Carrizzo, Albino et al. “New Nutraceutical Combination Reduces Blood Pressure and Improves Exercise Capacity in Hypertensive Patients Via a Nitric Oxide-Dependent Mechanism.” Journal of the American Heart Association vol. 9,5 (2020): e014923. doi:10.1161/JAHA.119.014923 ; Di Pietro, Paola et al. “A Novel Combination of High-Load Omega-3 Lysine Complex (AvailOm®) and Anthocyanins Exerts Beneficial Cardiovascular Effects.” Antioxidants (Basel, Switzerland) vol. 11,5 896. 30 Apr. 2022, doi:10.3390/antiox11050896 )

We thank the reviewer for the interesting comment. A new section entitled “6. ED therapeutic approach” has been added to the manuscript. Concerning the reviewer’s criticisms, the following sentences have been added in that section:

“Moreover, although large and numerous clinical trials in humans are missing, implementation of nutraceuticals before starting any type of treatment, or eventually in addition to it, may represent an important resource for ED prevention and maintenance of vascular health, since they may promote endothelial vasorelaxation, exert antioxidant properties and modulation of the lipidic profile. Furthermore, patient’s potential predisposition to start a natural substances based therapy could represent the best way to increase patient’s compliance to a preventive strategy and thus potentially reducing incidence of CVDs. In the end, it is reasonable to consider the use of nutraceuticals as complementary to the “traditional” lifestyle changes and “common” pharmacological therapies in order to improve the patient’s outcome.49-51”

Related references are the 49-51 as requested.

- The conclusion would benefit from added considerations. What should future studies search for in ED from Endothelial dysfunction and what do the authors think should the clinical recommendations suggest in patients with ED at risk for CVD.

We implemented our conclusions as requested. The following sentences have been added in the conclusions:

“Screening and diagnosing ED is essential for the primary and secondary prevention of CVDs, as the assessment of ED represents an easy and low-cost prognostic tool if compared to various cardiovascular biomarkers or more invasive exams. For men believed to have predominantly vasculogenic erectile dysfunction, initial cardiovascular risk stratification should be performed, for example applying the Framingham Risk Score. Management of men with erectile dysfunction who are at low risk for cardiovascular disease should focus on risk-factor control; whereas men at high risk, including mostly those with active cardiovascular symptoms, should be referred to a cardiologist before starting any kind of therapy for ED. Intermediate-risk men should undergo noninvasive evaluation for subclinical atherosclerosis and usually be re-stratified in any of two previous groups.”

Reviewer 2 Report

1. In the Abstract, the authors described that ED often precedes CVD, however, they also reported that several factors including DM, CVDs etc. are known risk factors for ED. Please clarify the temporality of the cause-effect between ED and CVD for this review.

2. Figure 1 should be revised and more specific for illustrating the potential mechanism of endothelial dysfunction and ED.

Author Response

Reviewer 2

  1. In the Abstract, the authors described that ED often precedes CVD, however, they also reported that several factors including DM, CVDs etc. are known risk factors for ED. Please clarify the temporality of the cause-effect between ED and CVD for this review.

We apologise but we are not sure of having understand the question of the reviewer. In the abstract we stated that ED and cardiovascular disease (CVD) should be considered two different manifestations of the same systemic disorder, with a shared aetiological factor being the endothelial dysfunction and with common risk factors. Furthermore, it was stated that penile arteries have a smaller size compared with coronary arteries; thus, for the same level of arteriopathy a more significant blood flow reduction will occur in erectile tissue compared with coronary circulation. As a result, ED often precedes CVD and its diagnosis offers a window of opportunity for cardiovascular risk mitigation.

As indicated in Figure 1, from the Ed to the CVD we have 2-5 years of time window to diagnose subclinical CVD and prevent future major events. We added this issue to the abstract hoping that this was the request of the reviewer. If not we are available to perform further revisions.

  1. Figure 1 should be revised and more specific for illustrating the potential mechanism of endothelial dysfunction and ED.

Figure 1 was elaborated and implemented with additional information. We hope that now it better illustrates the pathogenetic pathway from the healthy endothelium to the CVD. The aforementioned temporality of the cause effect between ED and CVD is highlighted in this figure too.

(Please see the attachment.)

Reviewer 3 Report

This brief review is interesting and well written.

This reviewer raises few issues that need to be addressed by authors in order to enrich the text.

1- Actually, some interesting mechanisms of endothelial dysfunction were observed. In particular, carbonic anhydrase-I induced by high glucose levels hampers endothelial cell permeability and determines endothelial cell apoptosis (J Am Heart Assoc. 2014 Apr; 3(2): e000434.  doi: 10.1161/JAHA.113.000434). Moreover, it has been described the role of TNF alpha in endothelial dysfunction in presence of high glucose. (Nutr Metab Cardiovasc Dis. 2007 May;17(4):274-9. doi: 10.1016/j.numecd.2005.11.014.) These issues and above references should be commented on.

2- Recently some interesting reviews have explained the potential endothelial protective role of drugs such as metformin (Biomedicines. 2020 Dec 22;9(1):3. doi: 10.3390/biomedicines9010003.) and SGLT2i (Biomedicines. 2021 Sep 29;9(10):1356. doi: 10.3390/biomedicines9101356.) which are now also used in non-diabetic patients. This important issue should be commented on with the above references.

Author Response

Reviewer 3

This brief review is interesting and well written.

This reviewer raises few issues that need to be addressed by authors in order to enrich the text.

  1. Actually, some interesting mechanisms of endothelial dysfunction were observed. In particular, carbonic anhydrase-I induced by high glucose levels hampers endothelial cell permeability and determines endothelial cell apoptosis (J Am Heart Assoc. 2014 Apr; 3(2): e000434.  doi: 10.1161/JAHA.113.000434). Moreover, it has been described the role of TNF alpha in endothelial dysfunction in presence of high glucose. (Nutr Metab Cardiovasc Dis. 2007 May;17(4):274-9. doi: 10.1016/j.numecd.2005.11.014.) These issues and above references should be commented on.

We thank the reviewer for his/her comment. Another paragraph was added in section 4 as follows:

“After having ascertained the role played by an altered NO release, it must be specified that this is just one of the many mechanisms involved in endothelial dysfunction. Other pathways worthy to be mentioned are the ones played by the Tumor Necrosis Factor alpha (TNF-a). It is produced by endothelial and smooth muscle cells in response to increased oxidative stress or inflammatory conditions.27-28 TNF-a induces ROS generation which may directly lower NO levels, apart from suppressing eNOS expression.29 Moreover, TNF-a in endothelial cells causes atherosclerosis30 and increases junctional cell permeability and ultimately cell apoptosis.31 Similar effects of altered cell permeability and increased cellular apoptosis where also described in diabetic patients, when studying carbonic anhydrase (CA) activity. CAs, a family of metalloenzymes involved also in pH regulation and ion transport, were shown to play a major role in diabetic microangiopathy. In patients with a high plasma glucose levels, CAs type I overexpression was responsible for altered endothelial cells permeability and consequent cellular death in vitro, which results in a decreased myocardial capillary density on histopathological specimens.32 Simultaneously CA type II was shown to be responsible for a particular ion exchanger (NHE-1) hyperactivity; which ultimately leads to ROS generation, myocardial dysfunction and cellular apoptosis.”

The requested references as well as others have been added (please see references 27-32).

2- Recently some interesting reviews have explained the potential endothelial protective role of drugs such as metformin (Biomedicines. 2020 Dec 22;9(1):3. doi: 10.3390/biomedicines9010003.) and SGLT2i (Biomedicines. 2021 Sep 29;9(10):1356. doi: 10.3390/biomedicines9101356.) which are now also used in non-diabetic patients. This important issue should be commented on with the above references.

A further section entitled “6. ED therapeutic approach” has been added to the manuscript. Concerning the reviewer’s criticisms, the following sentences have been added in that section:

“In fact, being an altered NO production due to endothelial dysfunction just one of the potential causes for ED, it should be reasonably feasible treating ED indirectly by treating its risk factors or predisposing conditions. New therapeutic approaches are being investigated; in particular oral hypoglycemic agents are among the ones showing the most promising results. It was already shown how, in patients affected by diabetes, specific medical treatments actually helped also in a lot of conditions that could be brought back to the primary disease. What is now catching the attention is that these drugs could provide benefits also to non-diabetic patients regardless of their glucose levels, hinting the presence of multiple mechanism leading to ED. Metformin, for example, was demonstrated to be able to reduce vascular inflammation, atherosclerosis and oxidative damage; regardless of changes in body weight or glycated hemoglobin.52 Employment of Gliflozins (another class of hypoglycemic agents) instead was correlated to a general improvement of the endothelial function, probably due to a reduce glucotoxicity, change in lipidic profile, anti-inflammatory action and also to a possible enhanced induced vasodilation.53,54”

Related references are 52-54.
